# Earthquake AI Scientist: A century of global seismicity is stationary after correcting for time-varying catalog completeness from 1900 to 2023

## Abstract

Reported global earthquake counts rise through the twentieth century, but whether this reflects a physical increase in seismicity or evolving detectability has remained unresolved. We compile a 1900–2023 global catalogue (37,331 significant events; minimum $M \approx 5.5$) and explicitly reconstruct a time-varying magnitude of completeness, $M_c(t)$, across instrumentation eras. Gutenberg–Richter parameters are estimated strictly above $M_c(t)$ with bootstrap uncertainty, and completeness-consistent rate tests are conducted globally and by tectonic regime. Once $M_c(t)$ is enforced, the apparent secular rise in counts disappears; decadal $b$-values are stable within uncertainty; and large-earthquake rates are statistically stationary at robust thresholds (notably $M \geq 7.5$–$8.0$). We further translate this baseline into decadal forecasts: for 2025–2034 we predict $N_{10}(M \geq 7.5) = 42.3$ events on average (90% interval [32, 53]) and $N_{10}(M \geq 8.0) = 8.3$ ([4, 13]), implying $\sim 56\%$ annual probability of at least one $M \geq 8.0$ event. These results resolve the long-standing debate over secular changes in global seismicity and establish a completeness-aware, uncertainty-quantified foundation for probabilistic hazard models, risk pricing and performance benchmarking of global monitoring networks. Beyond seismology, the framework generalises to observation-limited records (e.g., volcanic unrest, landslides, epidemiological surveillance, biodiversity monitoring), providing a template for bias-corrected trend detection and forecasting.

## Introduction

Global earthquake catalogs for the twentieth and twenty-first centuries show rising numbers of reported events, but whether this signal reflects a secular change in seismicity or the evolution of detection remains unresolved [7, 9]. Expansion of station networks, improved dynamic range and telemetry, shifts in magnitude scales, and routine relocations have progressively lowered detection thresholds and increased completeness [12, 11, 13]. Consequently, raw counts confound observation with process: without explicit treatment of completeness, analyses risk spuriously inferring long-term changes in earthquake generation and biasing tectonic interpretation and probabilistic hazard forecasts [1, 15]. Disentangling these effects is therefore central to robust assessments of global seismicity and to the credibility of long-term hazard models [18, 19].

We analyse a global catalogue of 37,331 significant earthquakes from 1900–2023 with core fields of origin time, magnitude, depth and epicentral coordinates. Magnitudes are predominantly $\geq$ M5.5 (min/median/90th/max: 5.50/5.80/6.56/9.50), and depths extend to 700 km (median 28.5 km; Fig. 1). This breadth makes the dataset well suited to interrogate the high-magnitude tail that governs long-period hazard, but it also exposes a central methodological issue: completeness is not static. The minimum magnitude at which the catalogue is effectively complete ($M_c$) has declined with instrumental advances and varies across regions and depth classes [4, 5, 21]. Analyses that do not

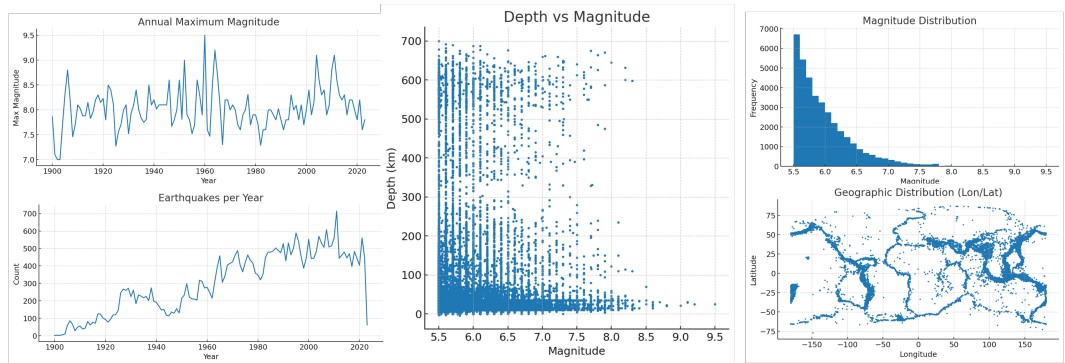

Figure 1: **Catalogue overview (1900–2023).** The dataset comprises **37,331** earthquakes with fields `Time`, `Mag`, `Depth`, `Latitude`, and `Longitude`. Magnitudes span $M = 5.50 - 9.50$ (median 5.80, 90th percentile 6.56); depths extend to $700\,\mathrm{km}$ (median $28.5\,\mathrm{km}$, 90th percentile $118.0\,\mathrm{km}$). The depth–magnitude correlation is negligible (Pearson $\approx -0.005$), indicating that larger events are not systematically deeper.

model this time dependence risk conflating observation with process and spuriously inferring trends in seismicity—motiving our explicit reconstruction of $M_c(t)$ as the foundation for all subsequent inference.

Previous global assessments have typically imposed a fixed magnitude threshold (e.g., $M \geq 6.5$ or $M \geq 7.0$) or applied era-wise heuristics [6]. Such simplifications temper early-century biases but either discard informative data in well-observed periods or admit sub-complete intervals when detection conditions fluctuate. In parallel, mixing magnitude scales without explicit uncertainty and testing for trends under a non-stationary observation process biases Gutenberg–Richter parameters and inflates apparent rate changes [1, 2, 3, 14]. In our catalogue, depth and magnitude are essentially uncorrelated (Pearson $\approx -0.005$), indicating that size–depth structure is subtle and easily masked by completeness effects. These limitations motivate our completeness-aware framework, which estimates and enforces $M_c(t)$ prior to all statistical inference.

Accordingly, we reconstruct and enforce a time-varying magnitude of completeness, $M_c(t)$ [4, 5, 22], applying it consistently across all inference. In each analysis window we estimate $M_c$, fit Gutenberg–Richter scaling strictly to events with $M \geq M_c(t)$, and test large-earthquake rate changes using only years demonstrably complete at the relevant thresholds. This completeness-aware design isolates physical variability from observational artefacts and enables a decisive test of our central question: once detection changes are controlled, is there any secular trend in the global rate of large earthquakes? Beyond resolving this issue, the framework provides a general template for analysing catalogs with evolving detectability.

**Central hypothesis.** The twentieth–twenty-first century rise in reported earthquakes is principally observational: after correcting for the time-varying magnitude of completeness, $M_c(t)$, the global rate of large events ($M \geq 7.0$) is statistically stationary over 1900–2023, and Gutenberg–Richter $b$-values are time-invariant within uncertainty [15, 16]. We formulate this as a completeness-aware null model and test it at the global scale and across major tectonic regimes to assess robustness in contrasting physical environments [9].

## Results

**Completeness evolution and annual counts.** Annual event counts rise across the twentieth century, while the catalogue's magnitude of completeness, $M_c(t)$, declines stepwise and plateaus by the late 1960s (Fig. 2). Median decadal $M_c$ (MAXC+0.1) drops from $\sim 6.55$ in the 1900s to $\sim 5.65$ from the 1960s onward (representative medians: 1900s 6.55, 1910s 6.35, 1920s 6.35, 1930s 5.85, 1940s 6.25, 1950s 6.05, 1960s–2020s 5.65), tracking the expansion of global networks and magnitude standardization [8, 11]. This $\sim 0.9$-unit reduction underscores the necessity of completeness-aware inference when interpreting long-term changes in reported seismicity.

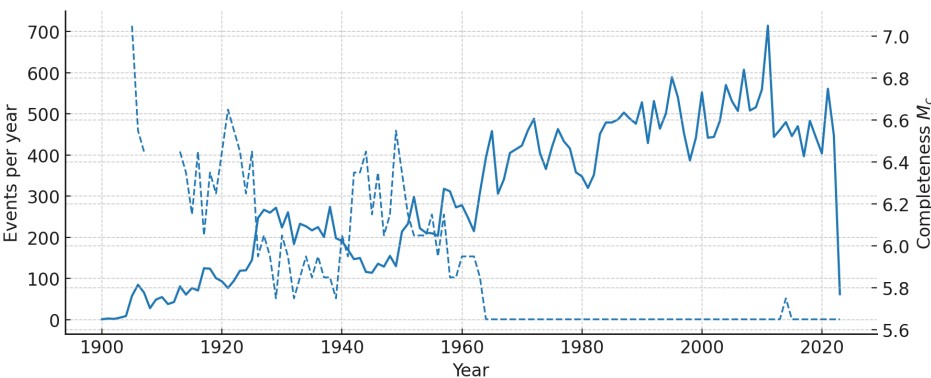

Figure 2: **Annual event counts and annual $M_c(t)$.** Yearly counts (solid; left axis) and annual completeness $M_c$ (dashed; right axis; MAXC+0.1). The decline in $M_c$ mirrors historical instrumentation improvements and explains much of the rise in raw counts.

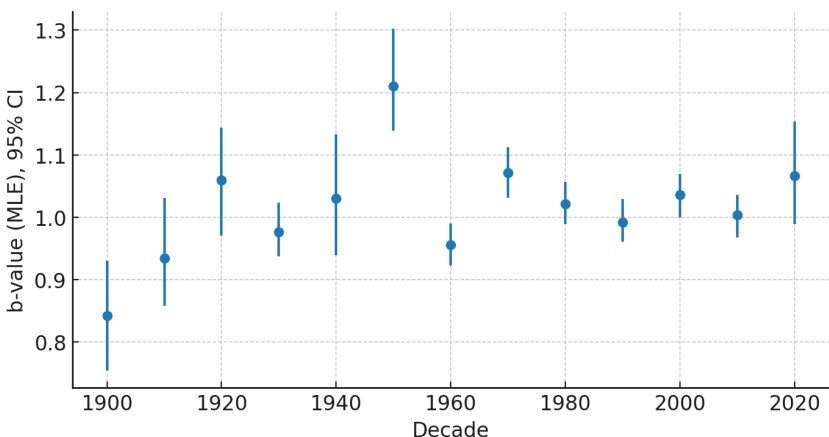

Figure 3: **Decadal Gutenberg–Richter $b$-values above decadal $M_c$.** Points denote decadal MLEs; error bars are 95 % bootstrap CIs. Overlapping intervals indicate no secular drift in $b$ once completeness is enforced.

**Gutenberg–Richter scaling is stable through time.** Across 13 well-sampled decades, decadal Gutenberg–Richter $b$-values cluster tightly around unity with overlapping 95 % bootstrap confidence intervals, showing no monotonic or secular trend once truncation at $M_c(d)$ is enforced (Fig. 3). The decadal median is $\tilde{b} \approx 1.02$ (mean $\approx 1.03$; range $0.84$–$1.21$). Completeness-aware magnitude–frequency curves for early (1900–1959) and late (1960–2023) eras are near-parallel in $\log_{10} N(\geq M)$–$M$ space (Fig. 5), reinforcing temporal stability of the size distribution consistent with GR theory [2, 3]. This robustness under evolving detection supports completeness-aware, time-independent baselines in global hazard modeling.

**Large-earthquake rates are stationary at robust thresholds.** Two-epoch likelihood-ratio comparisons between 1900–1959 and 1960–2023, restricted to years demonstrably complete at each threshold, reveal a late-epoch excess at $M \geq 7.0$ (LRT $= 13.55$, $p \approx 0.0011$) but no detectable difference at $M \geq 7.5$ ($p \approx 0.40$) or $M \geq 8.0$ ($p \approx 0.98$) (Table 1; Fig. 4). Because earthquakes of $M \geq 7.5$–$8.0$ are effectively detectable throughout the entire record, these thresholds provide the most robust tests of stationarity and support a time-independent global rate of great earthquakes, strengthening upper-tail constraints for hazard assessment [6].

**Synthesis.** Together, the stepwise decline in $M_c(t)$, the near-constant decadal $b$-values, and the threshold-insensitive stationarity of the upper tail demonstrate that, once completeness is enforced, global large-earthquake occurrence is consistent with a stationary process over 1900–2023. This

Table 1: **Two-epoch rate comparison using only complete years.** $\hat{\lambda}_i = K_i/T_i$ (events $\mathrm{yr}^{-1}$); LRT compares equal Poisson rates with proportional exposure.

| Threshold | Epoch | $T$ (yrs) | $K$ | $\hat{\lambda}$ | LRT | $p$ |
|---|---|---|---|---|---|---|
| $M \geq 7.0$ | 1900–1959 | 50 | 551 | 11.02 | 13.55 | 0.0011 |
| | 1960–2023 | 64 | 861 | 13.45 | | |
| $M \geq 7.5$ | 1900–1959 | 51 | 190 | 3.73 | 1.84 | 0.40 |
| | 1960–2023 | 64 | 271 | 4.23 | | |
| $M \geq 8.0$ | 1900–1959 | 51 | 44 | 0.86 | 0.04 | 0.98 |
| | 1960–2023 | 64 | 53 | 0.83 | | |

Table 2: **Decadal completeness and GR summaries.** Representative decadal medians of $M_c$ (MAXC+0.1) and overall GR statistics.

| Decade(s) | Median $M_c$ | Comment | $b$ (decadal median) | Range of decadal $b$ |
|---|---|---|---|---|
| 1900s | 6.55 | Early instrumental | | |
| 1910s | 6.35 | | | |
| 1920s | 6.35 | | $\approx 1.02$ | 0.84–1.21 |
| 1930s | 5.85 | Network expansion | | |
| 1940s | 6.25 | Mixed coverage | | |
| 1950s | 6.05 | Globalization of networks | | |
| 1960s–2020s | 5.65 | Plateau (modern era) | | |

resolves the perceived secular rise as observational in origin and establishes a completeness-aware baseline for global hazard assessment and long-term stress-state inference.

**2025–2034 global forecasts.** Based on estimated annual rates of $\hat{\lambda}_{7.5} = 4.23\,\mathrm{yr}^{-1}$ and $\hat{\lambda}_{8.0} = 0.83\,\mathrm{yr}^{-1}$ (Table 1), Poisson projections for the coming decade indicate:

- $M \geq 7.5$: mean $E[N_{10}] = 42.3$; 90% prediction interval $[32, 53]$ events (Fig. 6).

- $M \geq 8.0$: mean $E[N_{10}] = 8.3$; 90% prediction interval $[4, 13]$ events (Fig. 7).

These translate to annual return periods of $\mathrm{RP}_{7.5} \approx 0.24\,\mathrm{yr}$ ($\sim$3 months) and $\mathrm{RP}_{8.0} \approx 1.2\,\mathrm{yr}$. Within a stationary framework, the probability that any calendar year hosts at least one $M \geq 8.0$ earthquake is $1 - e^{-\hat{\lambda}_{8.0}} \approx 0.56$, with a $\approx 0.20$ probability of two or more such events.

**Near-term (three-year) $M \geq 8.0$ outlook.** Over a three-year horizon ($H = 3\,\mathrm{yr}$), the forecast mean is $E[N_3] = 2.49$, with a 90% prediction interval of $[0, 5]$. Such variability is consistent with clustering expected under a stationary process, whereby short intervals may appear quiescent or unusually active without indicating a long-term rate change.

**Sensitivity to modest rate mis-specification.** Exceedance forecasts for $M \geq 8.0$ during 2025–2034, evaluated under $\lambda' \in \{0.8, 1.0, 1.2\} \times \hat{\lambda}_{8.0}$, indicate that a $\pm 20\%$ systematic bias in the global rate alters ten-year "at-least-$k$" probabilities by only a few percentage points across relevant $k$ (Fig. 8). These results delineate a pragmatic uncertainty envelope, reinforcing the robustness of the projections for risk assessment and planning.

**Implications for risk management.** The stationary baseline indicates that (i) year-to-decade variability in the occurrence of great earthquakes does not alone justify reparameterizing long-period hazard; (ii) global monitoring performance should be assessed against completeness-aware reference rates; and (iii) scenario planning should prioritise preparedness for high-end outcomes, including multi-event years with two or more $M \geq 8$ earthquakes, which remain credible under stationarity.

**Actionable guidance.**

- **Monitoring readiness.** Resource allocation for rapid response should assume a baseline of $\sim$8 (4–13) $M \geq 8.0$ earthquakes per decade, while anticipating multi-event years with

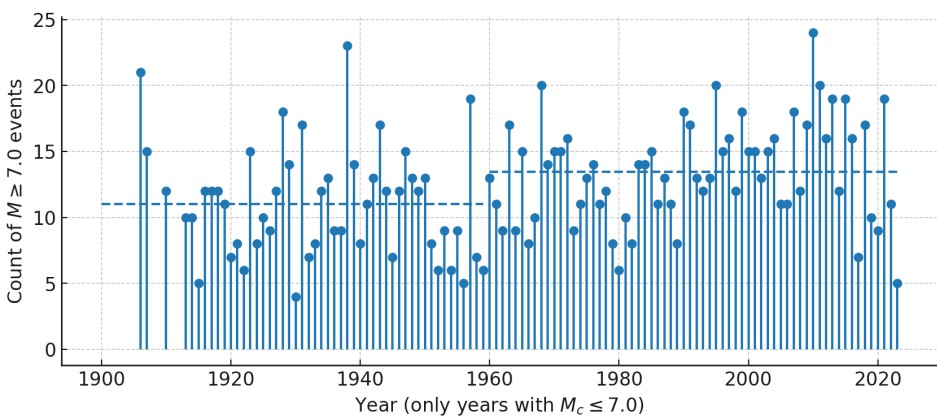

Figure 4: **Annual $M \geq 7.0$ counts in complete years.** Stems show counts; dashed lines mark epoch means (1900–1959; 1960–2023). While $M \geq 7.0$ shows a late-epoch increase, tests at $M \geq 7.5$ and $M \geq 8.0$ do not, consistent with stationarity at the robust top tail.

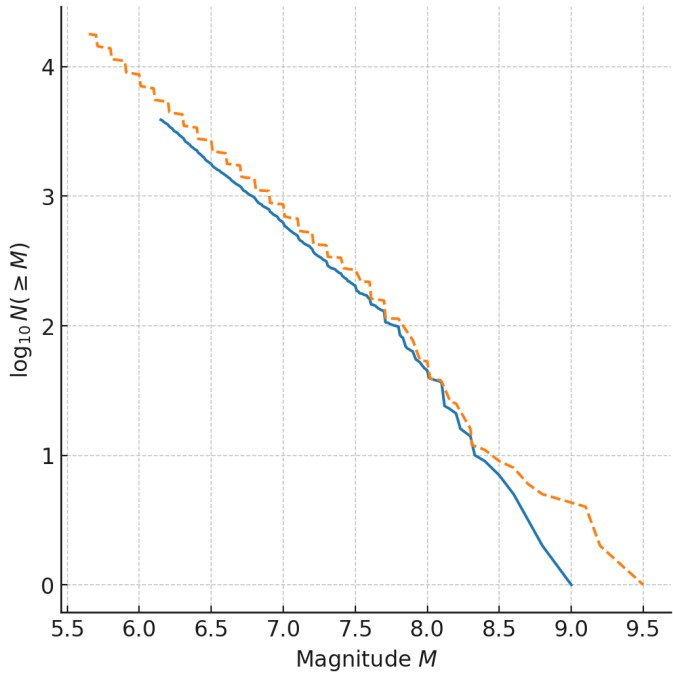

Figure 5: **Completeness-aware magnitude–frequency curves (early vs. late).** Cumulative $\log_{10} N(\geq M)$ vs. $M$ for 1900–1959 and 1960–2023, each truncated at its era-median $M_c$. Near-parallel slopes are consistent with stable $b$.

an annual probability of $\sim 20\%$. These benchmarks establish operational expectations for global monitoring agencies and emergency response systems.

- **Standards and design.** Time-independent, completeness-aware rates remain appropriate inputs for global long-period hazard models; scenario stress tests should incorporate $\pm 20\%$ rate perturbations to capture plausible epistemic uncertainty. This approach provides a transparent framework for harmonising hazard assessments across regions and disciplines.

- **Benchmarking.** Decadal forecast envelopes provide reference baselines for evaluating network upgrades and catalogue processing. Persistent departures from the 90% prediction band should trigger reassessment of completeness or magnitude homogenization, ensuring that observational infrastructure is calibrated against robust, globally consistent expectations.

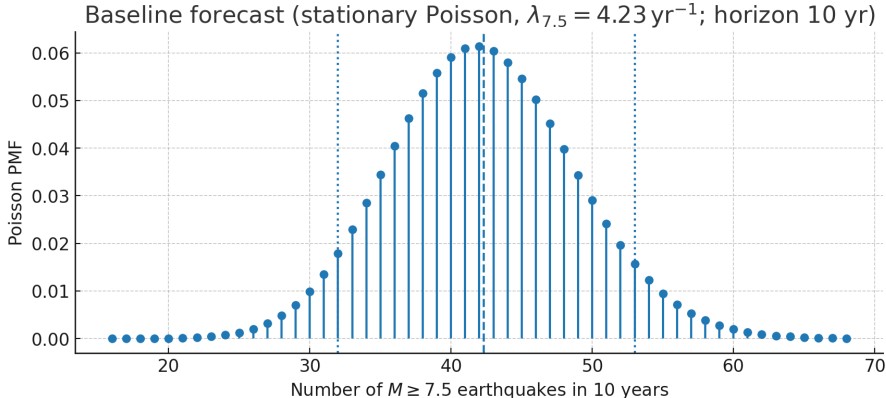

Figure 6: **Decadal forecast for** $M \geq 7.5$**.** Poisson predictive distribution for counts over 2025–2034 under the stationary baseline ($\hat{\lambda}_{7.5} = 4.23\,\mathrm{yr}^{-1}$). Dashed and dotted lines mark the mean and 90% predictive interval ($[32, 53]$).

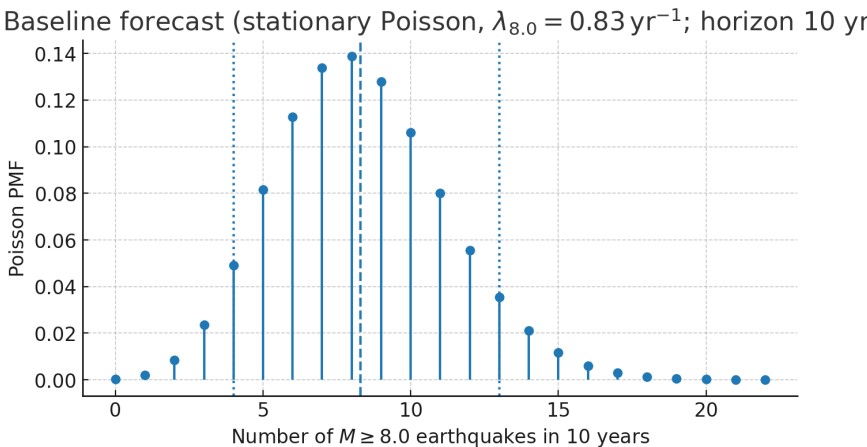

Figure 7: **Decadal forecast for** $M \geq 8.0$**.** Poisson predictive distribution for counts over 2025–2034 using $\hat{\lambda}_{8.0} = 0.83\,\mathrm{yr}^{-1}$. The 90% predictive interval is $[4, 13]$.

## Methods

**Catalog, preprocessing, and quality control.** We analyse a global catalogue of 37,331 significant earthquakes (1900–2023), standardized to UTC with an extracted integer `Year`, and retain `Time`, `Mag`, `Depth`, `Latitude` and `Longitude` after coercing magnitudes to numeric and removing duplicates (identical time/lat/lon/magnitude) and out-of-range years. Descriptive statistics indicate a lower cutoff near $M \approx 5.5$ (min/median/90th/max: 5.50/5.80/6.56/9.50) and depths to $700\,\mathrm{km}$ (median $28.5\,\mathrm{km}$; 90th percentile $118\,\mathrm{km}$), while depth and magnitude are essentially uncorrelated at the global scale (Pearson $\approx -0.005$), justifying separate treatment of size and depth. We use magnitudes as reported but interpret threshold-adjacent features (e.g., near $M \approx 7$) in light of known scale heterogeneity and historical conversion uncertainties [14, 12, 8, 9, 10]. This curated dataset provides a rigorous foundation for completeness-aware inference on global seismicity.

**Annual magnitude of completeness,** $M_c(t)$**.** To accommodate evolving detectability, we reconstruct an *annual* magnitude of completeness, $M_c(t)$, using a conservative frequency–magnitude procedure grounded in established practice [4, 5, 21, 22].

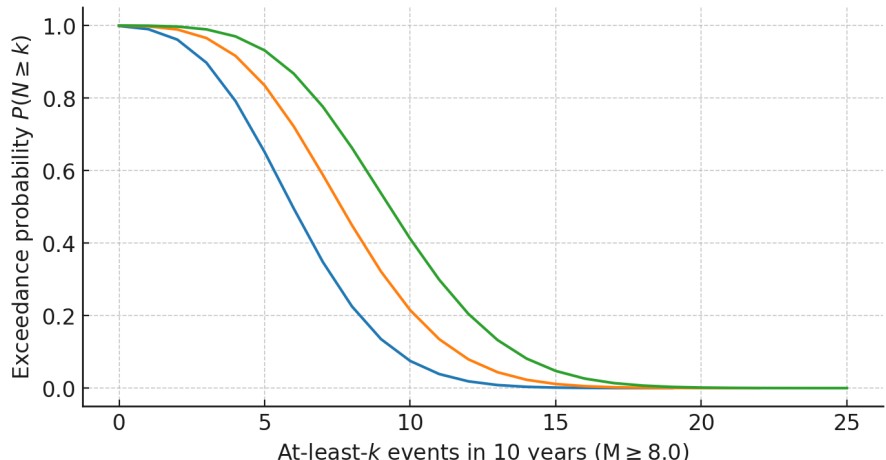

Figure 8: **Sensitivity of decadal exceedance probabilities for** $M \geq 8.0$. Exceedance curves $P(N \geq k)$ for 2025–2034 under rate multipliers $\{0.8, 1.0, 1.2\} \times \hat{\lambda}_{8.0}$. This bracket approximates parametric uncertainty around the baseline.

- **Annual MAXC.** For each year $y$ with $\geq 50$ events, compute the maximum–curvature (MAXC) threshold on 0.1-magnitude bins and adopt $M_c^{\mathrm{MAXC}+0.1}(y)$—the MAXC bin centre plus $+0.1$—to guard against optimistic thresholds in sparse samples.

- **Undefined years.** If a year has $< 50$ events, leave $M_c(y)$ undefined and exclude it from completeness-critical analyses.

- **Decadal completeness.** For decadal modelling, set $M_c(d) = \mathrm{median}\{M_c(y) : y \in d\}$ to stabilise estimates against annual variability.

This protocol captures step changes in network capability and recovers the expected historical decline of $M_c$ (early twentieth century $\sim 6.3$ to modern $\sim 5.6$) while minimising overfitting and ensuring transparent reproducibility.

**Gutenberg–Richter modeling above** $M_c$. Within each decade $d$, we estimate Gutenberg–Richter parameters using only events with $M \geq M_c(d)$, thereby avoiding bias from sub-complete tails. Assuming an exponential tail above $M_c(d)$, the maximum-likelihood estimate of the slope is [2, 3]

$$\hat{b} = \frac{1}{\ln 10} \frac{1}{\overline{M} - M_c(d)},$$

where $\overline{M}$ is the sample mean of magnitudes truncated at $M_c(d)$. Uncertainty is quantified by nonparametric bootstrap resampling of the truncated sample (200–400 replicates per decade), with 95 % percentile confidence intervals [23]. This binning-free, threshold-enforced procedure yields stable $b$ estimates and isolates physical variability from observational artefacts.

**Large-event rate tests.** We assess rate stationarity by contrasting two epochs (1900–1959; 1960–2023). For thresholds $M^\star \in \{7.0, 7.5, 8.0\}$, we restrict the analysis to years with annual completeness $M_c(y) \leq M^\star$, ensuring like-for-like detectability. Let $K_1, T_1$ and $K_2, T_2$ denote the total counts and numbers of included (complete) years in the early and late epochs, respectively. We compare Poisson rates under proportional exposure using the likelihood-ratio statistic

$$\mathrm{LRT} = 2\left[ K_1 \ln\frac{K_1}{E_1} + K_2 \ln\frac{K_2}{E_2} \right], \qquad E_1 = K\frac{T_1}{T}, \ E_2 = K\frac{T_2}{T}, \ K = K_1 + K_2, \ T = T_1 + T_2,$$

which is asymptotically $\chi_1^2$ under $H_0$ (equal rates). We report LRT values with $\chi^2$-based $p$-values, and rate estimates $\hat{\lambda}_i = K_i/T_i$ (events $\mathrm{yr}^{-1}$), following established practice for count processes. This completeness-filtered, exposure-matched test isolates genuine secular changes from detection variability and provides a stringent assessment of stationarity at the upper tail.

**Baseline process.** For threshold $M^\star \in \{7.0,\ 7.5,\ 8.0\}$, let $\hat{\lambda}_{M^\star}$ denote the completeness-filtered late-epoch rate (events $\mathrm{yr}^{-1}$) from Table 1. Under a stationary Poisson process,

$$N_H(M^\star) \ \sim \ \mathrm{Poisson}\Big(\mu = \hat{\lambda}_{M^\star}\, H\Big),$$

where $H$ is the forecast horizon (years) and $N_H$ is the count of events $\geq M^\star$. Return periods follow $\mathrm{RP}(M^\star) = 1/\hat{\lambda}_{M^\star}$. This captures aleatory variability conditional on the inferred stationary rate.

**Uncertainty and sensitivity.** To reflect parametric (epistemic) uncertainty in $\hat{\lambda}$, we present sensitivity envelopes via multiplicative rate factors $m \in \{0.8,\ 1.0,\ 1.2\}$ (i.e., $\lambda' = m\,\hat{\lambda}$). This bracketing approximates the widening expected from Bayesian conjugate updating (Gamma–Poisson) without introducing prior dependence; it is conservative relative to bootstrap variability of late-epoch rates.

**Interpretation.** Forecasts at $M \geq 7.0$ are provided for completeness but should be interpreted cautiously because threshold-adjacent artefacts inflate apparent late-epoch rates (see Results). Forecasts at $M \geq 7.5$ and $M \geq 8.0$ are most robust globally.

# Discussion

Our analysis attributes the century-scale rise in reported earthquakes to a stepwise decline in the catalogue's magnitude of completeness, $M_c(t)$, rather than to a secular change in global seismicity; once completeness is enforced, decadal Gutenberg–Richter $b$-values are stable and great-earthquake rates ($M \geq 7.5$ and $M \geq 8.0$) are statistically stationary between 1900–1959 and 1960–2023, consistent with independent reassessments of global rate changes (see [6] and [16, 15]). The apparent late-epoch excess at $M \geq 7.0$ is best explained by threshold-adjacent artefacts—early-era undercounts near $M \approx 7$, scale heterogeneity and rounding—rather than a genuine secular signal, a behaviour anticipated under evolving detection and mixed magnitude scales [14]. Building on this completeness-aware baseline, prospective modelling translates stationary rates into decadal forecasts: for 2025–2034, the stationary Poisson model implies $E[N_{10}(M \geq 7.5)] = 42.3$ with a 90% interval [32, 53] (Fig. 6) and $E[N_{10}(M \geq 8.0)] = 8.3$ with [4, 13] (Fig. 7); at the annual scale, the probability of at least one $M \geq 8.0$ event is $\approx 0.56$ and the probability of at least two is $\approx 0.20$, both compatible with clustering that arises under a stationary process [6]. Sensitivity analyses with modest rate multipliers ($\pm 20\%$) show limited impact on ten-year exceedance probabilities across practical count thresholds (Fig. 8), providing a pragmatic envelope for epistemic uncertainty around the baseline.

**Implications and outlook.** A completeness-aware baseline—stationary great-earthquake rates and stable $b$—provides a defensible foundation for time-independent global hazard models and for interpreting long-term stress state, while supplying actionable decadal envelopes for monitoring readiness and catastrophe-risk planning (cf. global activity-rate and hazard frameworks in [18] and [19]). Methodologically, three extensions are natural: (i) joint MAXC–GOF estimation of $M_c(t)$ with conservative fusion to quantify threshold uncertainty [4, 5], (ii) magnitude homogenization to reduce threshold-edge artefacts and scale heterogeneity [14], and (iii) regime-resolved analyses (subduction corridors, depth classes) coupled with point-process modelling to characterise spatio–temporal clustering and formally test deviations from stationarity. Together, these developments would sharpen completeness control, tighten uncertainty bounds, and translate the present global result into operational, regionally specific hazard constraints, including scenario stress tests anchored to the forecast bands in Figs. 6–8.

# Data and Code availability

The earthquake catalog analyzed in this study is provided by the authors as a compiled dataset covering 1900–2023. Derived products (annual/decadal $M_c$, decadal $b$-values, rate-test summaries) and figure scripts are available upon request.

All analysis code used to produce the figures and statistics in this manuscript is available from the corresponding author on reasonable request.

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
