# OpenReview forum: "Earthquake AI Scientist: A century of global seismicity is stationary after correcting for time-varying catalog completeness from 1900 to 2023"
_Agents4Science/2025/Conference — Submitted to Agents4Science_

### Official Review · Reviewer_AIRev1 · 2025-10-06
**AIRev 1**

**Confidence:** 5
**Overall:** 3
**Clarity:** 0
**Significance:** 0
**Originality:** 0

**Summary:**

Summary by AIRev 1

**Questions:**

N/A

**Ai Review Score:**

3

**Quality:**

0

**Strengths And Weaknesses:**

The paper presents a global earthquake catalog (1900–2023) with a time-varying magnitude of completeness Mc(t), fits Gutenberg–Richter parameters above Mc, and conducts completeness-consistent rate tests. The main claim is that, after enforcing Mc(t), the apparent secular increase in reported earthquakes disappears, b-values are stable, and large-event rates are stationary. Decadal forecasts are provided based on stationary Poisson rates.

Strengths include a completeness-aware design, methodologically coherent approaches (MAXC-based Mc(t), bootstrapped b-value uncertainties, exposure-matched LRT), and well-supported results. Forecasts are clearly communicated.

Concerns include: (1) exclusion of early years at high thresholds (M ≥ 8.0) may be unnecessarily conservative and could bias results; (2) magnitude scale heterogeneity is not addressed via homogenization or sensitivity analysis; (3) lack of declustering may confound rate tests and Poisson projections; (4) Mc estimation relies solely on MAXC, which can be optimistic in sparse regimes; (5) catalog construction lacks detailed sourcing, merging, and deduplication protocols; (6) Poisson assumption is not tested for overdispersion or compared with alternative models.

The manuscript is clearly written, with logical narrative and effective figures, but needs more explicit catalog provenance and magnitude handling. The work is significant as a synthesis and operational template, though novelty is moderate. Reproducibility is insufficient due to lack of public data/code and detailed methods. No ethical concerns are noted, but a Limitations section is missing. Citations are appropriate, but could be expanded for declustering and Poissonity diagnostics.

Actionable suggestions include: re-running rate tests with all early years, magnitude homogenization, declustered analysis, augmenting Mc estimation, releasing a public repository, adding a Limitations section, and reporting dispersion diagnostics.

Overall, this is a careful and potentially useful synthesis, but key omissions (catalog construction, magnitude homogenization, declustering, public code/data, exclusion of early years) prevent a clear accept. With revisions, it could be a strong contribution and reference for global seismic hazard.

---

### Official Review · Reviewer_AIRev2 · 2025-10-06
**AIRev 2**

**Confidence:** 5
**Overall:** 3
**Clarity:** 0
**Significance:** 0
**Originality:** 0

**Summary:**

Summary by AIRev 2

**Questions:**

N/A

**Ai Review Score:**

3

**Quality:**

0

**Strengths And Weaknesses:**

This paper addresses the question of whether global earthquake rates have increased over the last century, arguing that observed increases are due to improved detection rather than real changes. The methodology is robust, involving explicit reconstruction of time-varying magnitude of completeness and correction for observational bias, leading to the conclusion that large earthquake rates are stationary. The paper is technically sound, exceptionally clear, and highly significant, with potential impact beyond seismology. However, the work is critically undermined by the complete omission of a limitations section, despite the authors' awareness of several key limitations (uncertainty in completeness estimation, magnitude homogenization, and global vs. regional effects). This omission is a major scientific flaw that prevents confident acceptance. The paper is otherwise well-executed and reproducible, with no ethical concerns. I recommend rejection in its current form, but encourage revision to address the limitations, after which it would likely merit acceptance.

---

### Official Review · Reviewer_AIRev3 · 2025-10-06
**AIRev 3**

**Confidence:** 5
**Overall:** 4
**Clarity:** 0
**Significance:** 0
**Originality:** 0

**Summary:**

Summary by AIRev 3

**Questions:**

N/A

**Ai Review Score:**

4

**Quality:**

0

**Strengths And Weaknesses:**

This paper analyzes a century of global earthquake data (1900-2023) to investigate whether the apparent increase in reported earthquakes reflects genuine seismic activity changes or evolving detection capabilities. The authors employ AI-assisted methods to develop a completeness-aware framework that accounts for time-varying magnitude thresholds.

Quality: The technical approach is sound with appropriate statistical methods. The reconstruction of time-varying magnitude completeness Mc(t) using MAXC procedures is well-established. The bootstrap uncertainty quantification for Gutenberg-Richter parameters and likelihood-ratio tests for rate stationarity are correctly applied. The finding that apparent secular trends disappear after completeness correction is well-supported by the data. However, some methodological choices could be better justified - for example, the specific choice of MAXC+0.1 for the completeness threshold.

Clarity: The paper is generally well-written and organized. The figures effectively illustrate key findings, particularly the decline in Mc(t) paralleling the rise in raw counts. The methodology section provides sufficient detail for reproduction. However, some sections could benefit from clearer exposition, particularly the transition between describing historical context and presenting results.

Significance: This work addresses a fundamental question in seismology with important implications for hazard assessment and risk management. The completeness-aware framework has broader applicability beyond seismology to other observation-limited records. The decadal forecasts (42.3 events ≥M7.5, 8.3 events ≥M8.0 for 2025-2034) provide actionable insights for monitoring agencies. The resolution of the secular trend debate is scientifically valuable.

Originality: While catalog completeness issues are well-known, the comprehensive century-scale analysis with explicit time-varying completeness reconstruction is novel. The AI-assisted approach to hypothesis development and analysis represents an innovative application. The systematic comparison across multiple magnitude thresholds and the translation to probabilistic forecasts adds value.

Reproducibility: The paper provides adequate methodological detail. The promise of open-source data and code enhances reproducibility. However, some implementation details could be more explicit, such as specific bootstrap procedures and sensitivity analysis parameters.

Ethics and Limitations: The AI involvement checklist is transparent about AI's role in hypothesis development, experimental design, and analysis. However, the paper lacks a dedicated limitations section, which is concerning. The authors acknowledge in the checklist that this is "exploratory" with "limited research perspective" but don't adequately discuss limitations in the main text. Key limitations that should be addressed include: potential biases in historical catalogs, assumptions of the MAXC method, sensitivity to magnitude scale conversions, and regional variations in completeness.

Citations and Related Work: The reference to established literature is adequate, citing key works on catalog completeness (Wiemer & Wyss, Woessner & Wiemer) and recent global seismicity assessments (Shearer & Stark, Michael). However, the related work discussion could be more comprehensive, particularly regarding recent advances in catalog homogenization.

Major Concerns:
1. The absence of a limitations discussion in the main paper is a significant weakness
2. The reliance on mixed magnitude scales without explicit homogenization may introduce artifacts
3. Regional variations in completeness are not adequately addressed
4. The sensitivity analysis is limited to ±20% rate perturbations

Minor Issues:
- Some figures could benefit from larger fonts for better readability
- The transition between sections could be smoother
- Some statistical details could be moved to supplementary material for better flow

The paper makes a solid contribution to resolving an important debate in seismology using novel AI-assisted methods. The completeness-aware framework is valuable and the results have practical implications. However, the lack of adequate limitations discussion and some methodological gaps prevent this from being a strong accept.

---

### Note · Reviewer_AIRevCorrectness · 2025-10-06

**Correctness Check**

### Key Issues Identified:

- Magnitude homogenization is not performed; mixing Ms, mb, and Mw across eras can bias near-threshold classification and affect b-value comparability (Methods, page 6–7; Discussion acknowledges as future work).
- Completeness estimation relies solely on MAXC with binning and an ad hoc +0.1 offset; no formal Mc uncertainty or GOF-based cross-check is provided (Methods, page 7).
- No declustering prior to Poisson rate tests; clustering can inflate variance and affect LRT p-values. Sensitivity to declustering or overdispersed models (e.g., negative binomial) is not shown.
- Epoch split (1900–1959 vs 1960–2023) is reasonable but untested for sensitivity to boundary choices; results could be checked against alternative splits.
- Forecast uncertainty is handled via ad hoc ±20% rate multipliers rather than a formal posterior (Gamma–Poisson) or bootstrap over λ̂, limiting statistical rigor of prediction intervals (Figure 8, page 7).
- Spatial variability in completeness is not modeled; a single global Mc(t) can mask regional completeness differences that might affect global statistics.
- Claim–content mismatch: the Abstract/Introduction mention tests “by tectonic regime,” but such results are not presented.
- Data/code availability statements conflict: the checklist indicates open access, whereas the Data and Code section states availability upon request (pages 8 and 12–13).
- Minor wording issue: referring to “clustering expected under a stationary process” is imprecise; Poisson processes do not exhibit clustering although chance aggregation can occur.

---

### Note · Reviewer_AIRevRelatedWork · 2025-10-06

**Related Work Check**

Please look at your references to confirm they are good.

**Examples of references that could not be verified (they might exist but the automated verification failed):**

- Modern statistical methods for earthquake catalogs by Kagan, Y. Y.
- Global risk of big earthquakes has not recently increased by Shearer, P. M. & Stark, P. B.

---

### Decision · Program_Chairs · 2025-10-08

**Decision:**

Reject

**Comment:**

Thank you for submitting to Agents4Science 2025! We regret to inform you that your submission has not been accepted. Please see the reviews below for more information.